# Mechanical and Electrical Interaction of Biological Membranes with Nanoparticles and Nanostructured Surfaces

**DOI:** 10.3390/membranes11070533

**Published:** 2021-07-14

**Authors:** Jeel Raval, Ekaterina Gongadze, Metka Benčina, Ita Junkar, Niharika Rawat, Luka Mesarec, Veronika Kralj-Iglič, Wojciech Góźdź, Aleš Iglič

**Affiliations:** 1Group of Physical Chemistry of Complex Systems, Institute of Physical Chemistry, Polish Academy of Sciences, 01-224 Warsaw, Poland; jlraval@gmail.com (J.R.); wtg@ichf.edu.pl (W.G.); 2Laboratory of Physics, Faculty of Electrical Engineering, University of Ljubljana, 1000 Ljubljana, Slovenia; ekaterina.gongadze@fe.uni-lj.si (E.G.); niharika.rawat@fe.uni-lj.si (N.R.); mesarec.luka@gmail.com (L.M.); 3Department of Surface Engineering and Optoelectronics, Jožef Stefan Institute, 1000 Ljubljana, Slovenia; metka.bencina@ijs.si (M.B.); ita.junkar@ijs.si (I.J.); 4Laboratory of Clinical Biophysics, Faculty of Health Sciences, University of Ljubljana, 1000 Ljubljana, Slovenia; kraljiglic@gmail.com; 5Laboratory of Clinical Biophysics, Chair of Orthopaedics, Faculty of Medicine, University of Ljubljana, 1000 Ljubljana, Slovenia

**Keywords:** lipid bilayer electrostatics, zwitterionic lipid bilayers, electric double layer, osmotic pressure, orientational degree of freedom of lipid headgroups, orientational ordering of water dipoles, adhesion of lipid vesicles, lipid bilayer elasticity, lipid vesicle shapes

## Abstract

In this review paper, we theoretically explain the origin of electrostatic interactions between lipid bilayers and charged solid surfaces using a statistical mechanics approach, where the orientational degree of freedom of lipid head groups and the orientational ordering of the water dipoles are considered. Within the modified Langevin Poisson–Boltzmann model of an electric double layer, we derived an analytical expression for the osmotic pressure between the planar zwitterionic lipid bilayer and charged solid planar surface. We also show that the electrostatic interaction between the zwitterionic lipid head groups of the proximal leaflet and the negatively charged solid surface is accompanied with a more perpendicular average orientation of the lipid head-groups. We further highlight the important role of the surfaces’ nanostructured topography in their interactions with biological material. As an example of nanostructured surfaces, we describe the synthesis of TiO_2_ nanotubular and octahedral surfaces by using the electrochemical anodization method and hydrothermal method, respectively. The physical and chemical properties of these nanostructured surfaces are described in order to elucidate the influence of the surface topography and other physical properties on the behavior of human cells adhered to TiO_2_ nanostructured surfaces. In the last part of the paper, we theoretically explain the interplay of elastic and adhesive contributions to the adsorption of lipid vesicles on the solid surfaces. We show the numerically predicted shapes of adhered lipid vesicles corresponding to the minimum of the membrane free energy to describe the influence of the vesicle size, bending modulus, and adhesion strength on the adhesion of lipid vesicles on solid charged surfaces.

## 1. Introduction

Biological membranes are an essential constituent of living cells. Their main role is to separate the interior of the cell from its surroundings, allowing for selective transport of specific materials across the membrane [1]. This article focuses on the interaction of biological membranes with nanostructured surfaces and nanoparticles [1,2,3]. The main building block of the biological membranes is the lipid bilayer with embedded inclusions such as proteins and glycolipids. Isotropic and anisotropic membrane proteins may induce local changes in the membrane curvature [1,4,5], often resulting in global changes in the cell shape [6,7,8,9,10,11,12]. The nonhomogeneous lateral distribution and the phase separation of membrane inclusions (nanodomains) can induce local changes in the membrane curvature and are, therefore, the driving force for transformations of the cell shape [6,7,8,9,12,13,14]. The biological and lipid membranes possess some degree of in-plane orientational ordering [1,7,8,10,15,16,17,18], including the nematic type of ordering [19,20,21], which are also important for the stability of different membrane shapes.

The configuration and shape changes of membranes are, in general, correlated with many important biological processes [1,6,13,22]. The shapes of cells are also influenced by the membrane skeleton and cytoskeleton forces [1,11,13,22,23,24,25,26,27,28]. Among them, the ATP consuming forces of the membrane skeleton and cytoskeleton are of major importance for sustaining different cell functions [11,12,22,28,29]. Consequently, new theoretical approaches for modeling these cell shape changes under the influence of energy-consuming active forces have been developed recently [11,12,28,29].

The focus of this paper (partially a mini review) is the interaction of nanoparticles (NPs) and nanostructured solid surfaces with cell membranes and lipid bilayers. Certain aspects of membrane–solid surface electrostatics and adhesive interactions are elucidated too.

## 2. Interaction of Nanoparticles with Cell Membrane

The shape and biological functions of membranes can be strongly influenced by attached, encapsulated (Figure 1) [30], or intercalated inclusions such as nanoparticles (NPs). Unique optical, electronic, catalytic, and magnetic properties of NPs make them very interesting for a variety of biomedical applications [1,31,32,33,34]. For example, functional NPs and quantum dots are potential candidates for drug delivery, as well as carriers for cancer therapy [33,35]. When NPs interact with cells, the first barrier that NPs encounter is the plasma membrane. Intra- and extracellular transport of NPs are possible by a dynamic membrane shape transformation that involves a change in the membrane curvature (encapsulation) [30,32,33,36] (Figure 1). Membrane deformations may progress passively, that is, without employing an additional energy source, driven solely by the interaction between the membrane and NPs. Viral budding comprises such an example [37]. The intracellular entry of genetic material is presently receiving considerable attention due to the COVID-19 crisis. Electrostatic interactions [38,39] may facilitate NP or virus internalization via encapsulation [30,33] (Figure 1). Understanding the interplay between the membrane elastic and electrostatic properties of the NP–membrane complex, toward the encapsulation of NPs by the cell membrane, is also relevant for cellular drug uptake, viral budding, biotechnological applications, and studying the interactions of inorganic NPs with biological membranes [32,33].

Another possible interaction of NPs with the membrane is the attachment (adsorption) of the former on the membrane surface [34,38,40], encapsulation [30], or their intercalation in the membrane [41,42,43]. The resulting configuration could be driven by the NP shape, charge, size, and stiffness; it also depends on the nature of the NP–membrane interaction [30,34,38,39,41,44]. For example, hydrophobic or cationic NPs with diameters smaller than 5 nm can be successfully embedded within the membrane bilayer. On the other hand, anionic NPs of the same size or larger NPs can only interact with the outer surface of the membrane [45]. NPs interacting with membranes may induce lateral tension that results in pore formation, either transient or permanent; the pores are actually stabilized by NPs. Nanoparticles could also cluster within the membrane, and the resulting change in the membrane mechanics could significantly influence its biological function and could even result in membrane disruption [46]. Simulations demonstrated that the properties of both the membrane and the NPs are equally important in explaining the membrane uptake of the latter [30,41].

## 3. Interaction of Cells with Nanostructured Surfaces

Apart from exploiting the NP–membrane interactions in biomedical applications, the applications of micro- and nano-structured surfaces (Figure 2) in biomedicine have also attracted significant attention. Among them, of particular interest is the design of nanotopographic features such as biomimetic interfaces for implantable devices [47,48,49]. In vitro results have shown that the surface features on the nanometer scale stimulate and control several molecular and cellular events on tissue/implant interfaces, which can be observed by differences in the cell morphology, orientation, cytoskeleton organization, proliferation, and gene expression [47,48,49]. In the last decade, TiO_2_ nanotubes fabricated via electrochemical anodization (Figure 3 and Figure 4) have attracted significant attention toward medical applications [1,50].

Among others, nanotopography on TiO_2_ surfaces exhibits reduced activation and the aggregation of platelets [52], and the controlled adhesion and proliferation of endothelial and smooth muscle cells [3]. The excellent potential of TiO_2_ nanotubes in medicine and biotechnology is mainly due to their high effective surface area, increased surface charged density [47,48], and the possibility to vary their geometry (diameter and length), which could be specially designed/adapted for a desired biological response (cell selectivity). TiO_2_ nanotubes have also been shown to increase selective protein adsorption [53] and, thus, they enhance the biological response. For example, studies have shown that TiO_2_ nanotubes increase bone growth/regeneration, are antibacterial, and reduce inflammation [54,55]. Moreover, the endothelium formed on the surfaces with nanoscale topography exhibits an enhanced expression of anti-thrombogenic genes, providing a more extended coagulation cascade, probably due to a thicker oxide layer and specific topography [49].

Several reports have shown that nanotopography significantly influences cell behaviors, i.e., adhesion, proliferation, and differentiation [47,56,57]. It was recently shown that the specific nanotopography of TiO_2_ used for cardiovascular stents can improve their bio-/hemo-compatibility via the increased adhesion and growth of human coronary artery endothelial cells and reduced adhesion and activation of platelets [3]. The surface nanostructurization of titanium (specifically the formation of TiO_2_ nanotubes with different diameter) was shown to alter physicochemical properties, such as wettability [2] and surface chemistry, which consequently affect the interactions of TiO_2_ nanostructured surfaces with cells [3,47,58]. In vivo studies have shown improved endothelization and reduced neointimal thickening on nanostructured stents compared to bare-metal stents [58]. Moreover, the study performed by Peng et al. [59] showed that the TiO_2_ nanotubular surface significantly enhances endothelial cell proliferation, while, at the same time, the growth of vascular smooth muscle cells is reduced.

**Figure 4 membranes-11-00533-f004:**
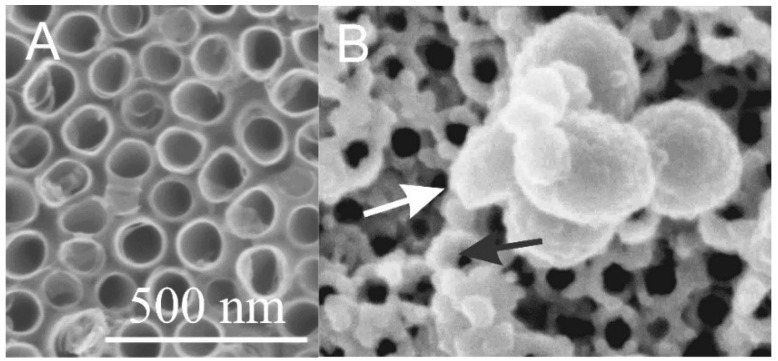
Scanning electron microscope (SEM) images of TiO_2_ nanotubular surface (**A**) and the TiO_2_ nanotubular surface that was exposed to the isolate of extracellular vesicles (white arrow) from blood (**B**). Material deposited also on the top surface of the TiO_2_ nanotube walls is denoted by a black arrow. Adapted with permission from [60,61].

Surface nanotopography may thus strongly influence the adhesion and proliferation of cells [47]. Experimental studies of the interaction between lipid bilayers and solid surfaces have indicated that the specific structure of the surface can induce even the phase transitions in the membrane upon its adhesion to the surface [62].

Among TiO_2_ nanostructured surfaces presented in Figure 3, TiO_2_ nanotubular surfaces were synthesized by electrochemical anodization [3,48,53,63], while hydrothermal treatment was employed for synthesizing other structures. To illustrate the interaction of nanostructured surfaces with biological systems, two nanostructuring procedures—electrochemical anodization and hydrothermal treatment—were used for the synthesis of TiO_2_ nanotubes (Figure 3b) and octahedral nanostructures (Figure 3d), respectively. In the case of electrochemical anodization, nanotubes with a diameter of 100 nm were synthesized by using HF and ethylene gylcol as electrolytes (described in detail in Refs. [60,61]), whereas for the hydrothermal method, Ti foils were exposed to a basic medium of Ti isopropoxide suspension. Briefly, in the hydrothermal method, Ti foil (0.1 mm, 99.6+%, Advent) was washed in acetone, ethanol, and water (in each for 5 min). Afterward, the samples were dried at 70 °C in the furnace. Ti isopropoxide (1 mL, 99.999% trace metals basis, Sigma Aldrich) was used as an ion precursor, and KOH (flake, 85%, Alfa Aesar) was added to the solution until the pH reached 10. The cleaned and dried Ti foil was placed at the bottom of a Teflon vessel in a hydrothermal reactor and poured with Ti isopropoxide suspension. The reaction was carried out at 200 °C for 24 h. Samples were then vigorously washed with deionized H_2_O, dried under the stream of N_2_, and used for further experiments.

As-prepared nanostructured surfaces were used to study interactions with blood platelets to evaluate the materials’ potential thrombogenic nature as in [3]. It is of tremendous importance that metal implants, used as vascular stents, inhibit or decrease potentially fatal thrombosis and restenosis conditions. The stent surface should prevent excessive adhesion and aggregation of platelets as it can lead to blood clot formation/thrombosis. The adhesion and activation of platelets on the surface presented in Figure 5 is an indicator of how hemocompatible a material is; the lower platelet adhesion and activation, the higher is material compatibility with blood (hemocompatibility) [64,65]. It has already been recognized that protein adsorption and its conformation play an important role in platelet adhesion. Fibrinogen adsorption has been so far recognized as an important factor determining platelet adhesion and activation [66]. Studies have already shown that surface nanotopography may significantly influence platelet adhesion and activation, mainly due to nanotopography-induced changes in fibrinogen adsorption [66]. The study showed that the confirmation of adsorbed fibrinogen is highly linked to nanotopographic features of the surface. In the present study, other surface features such as wettability, roughness, and chemistry remained unchanged. However, when designing nanomaterials, it is sometimes hard to eliminate other surface features, which may even change over time. It was shown that TiO_2_ nanotubular surfaces (Figure 3) tend to age with time and lose their hydrophilic characteristic [2]. The altered surface wettability and slight changes in chemistry significantly influence platelet adhesion. In this case, a decrease in the wettability of nanotubes 15, 50, and 100 nm in diameter resulted in a higher platelet adhesion and activation compared to more hydrophilic nanotubes [3]. However, the optimal surface nanostructure may significantly influence the biological response when other surface features are optimized.

In Figure 6, the scanning electron microscopy (SEM) images of plain Ti foil, TiO_2_ nanotubes, and hydrothermally treated Ti foil are presented. TiO_2_ nanotubes prepared by electrochemical anodization are 100 nm in diameter, while the nanostructured surface of hydrothermally treated Ti foil consists of octahedral nanoparticles with sizes of about 150–300 nm. More detailed information about surface roughness can be obtained from images taken via atomic force microscopy (AFM). The 3D image of surface topography obtained from AFM reveals that plain Ti foil has no special surface morphology, and the estimated surface roughness (Ra) evaluated from the images is about 11.7 nm. However, both nanostructured surfaces have a higher surface roughness (Ra), about 40 and 49 nm for the TiO_2_ nanotubular and hydrothermal surface, respectively. It should be mentioned that in the case of nanotubes, the ability of the AFM tip to enter the hollow interior of the nanotube is limited; however, it gives us some additional information about surface topography, especially the variation in the height of nanotubes. The evaluated difference in height between nanotubes is about 190 nm. In the case of hydrothermal treatment, the difference in the height of octahedral particles is very similar, about 185 nm. Thus, the main difference between both surfaces is in the width and shape of the nano-features.

Surface chemistry should also be considered when designing the biomaterial surface. Using the XPS technique, it is possible to determine the chemical composition of the elements on the top surface (about 5 nm in depth), which interact with the biological environment. The plain, as well as the nanostructured, surfaces were analyzed by X-ray photoelectron spectroscopy (XPS). The results from XPS analysis are presented in (Table 1). An increase in titanium and oxygen concentration after both hydrothermal and anodization processes is observed compared to the control (Ti foil), partially also due to the removal of surface hydrocarbon contamination (lower carbon content). In the case of the anodization process, fluorine is also detected on the surface due to HF used as an electrolyte. Thus, after altering the surface nanotopography, changes in surface chemistry should also be considered, as they may influence biological interactions. As the wettability of surfaces is also highly correlated with biological response, it should be mentioned that both freshly synthesized nanostructured (nanotubular and octahedral) surfaces are hydrophilic with water contact angle (WCA) below 5°, while the WCA measured on plain Ti foil exhibits a more hydrophobic characteristic (WCA of about 75°). A detailed description of wettability studies can be found in Ref. [3].

After analyzing the surface properties, the interaction of surfaces with the biological environment should be studied. For the application of biomaterial for vascular implants, the interaction with whole blood should be evaluated. According to Goodman et al. [64], platelet adhesion and activation can be evaluated from their shape. The activation of platelets can be described in three steps: adhesion, spreading/aggregation of platelets, and activation of platelets/formation of a thrombus clot [67]. The morphology of adherent platelets is commonly described as round (R), dendritic (D), spreading dendritic (SD), spreading (S), and fully spreading (FS). Platelets with F and FS are considered activated platelets (Figure 5) [64,68].

**Figure 5 membranes-11-00533-f005:**
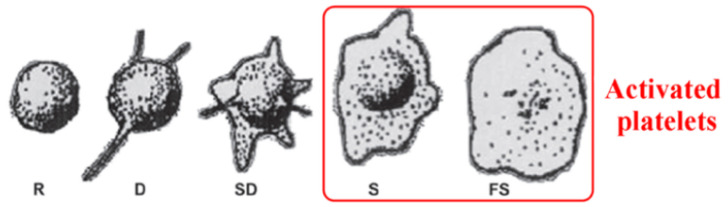
Platelet’s morphology after the adhesion. Their shape is marked as: round (R), dendritic (D), spreading dendritic (SD), spreading (S), and fully spread (FS). Reprinted with permission from Ref. [68] copyright 2014 American Chemical Society.

The interaction of whole blood and Ti surfaces is presented in Figure 6. On the smooth, non-treated Ti foil surface (control), the platelets are preferentially in dendritic form, and platelet-to-platelet adhesion (aggregation) is observed (Figure 6g). Platelets are aggregated and activated as the specific platelet shape change is observed, i.e., the multiple filopodial extensions from the platelet body and lamellipodia formation can be seen (SD and S shape). On the surface of the TiO_2_ nanotubular layer, the platelets are extensively spread (S and FS shape), agglomerated, and express lamellipodia and numerous filopodia (Figure 6h). The area of spreading is significantly higher than on the surface of Ti foil.

On the surface of hydrothermally treated Ti foil, individual discoid/dendritic platelets with no obvious pseudopods are observed. Platelets are mainly in the round (R) and dendritic (D) form, indicating low surface interaction. Moreover, compared to the TiO_2_ nanotubular layer (Figure 6h), a lower number of platelets is detected on these surfaces (Figure 6i), and platelets seem to not be in the activated form. The results presented in this section suggest that nanostructured surfaces may reduce or even increase platelet adhesion and activation. Although both nanostructured surfaces (i.e., nanotubular and octahedral) have similar surface chemistry (higher concentration of Ti and O atoms, with the addition of F for the case of the nanotubular layer) and wettability (hydrophilic) compared to plain Ti foil, their interaction with platelets seems to differ significantly in attachment and proliferation. Thus, it is important to note that a specific surface nanotopography may significantly affect the surface interaction with platelets. In Refs. [64,68], it was also shown that platelets, endothelial cells, and smooth muscle cells selectively interact with the TiO_2_ nanotubes with various diameters, which, again, implies that surface nanotopography plays a significant role in the adhesion of biological material. Hence, by modifying the surface and with the appropriate surface chemistry, the influence of surface hydrothermally treated Ti can potentially prevent thrombosis, which could, to some extent, be correlated with its specific nanotopographic features, as the surface wettability, as well as chemistry, of both nanostructured materials was shown to be very similar.

**Figure 6 membranes-11-00533-f006:**
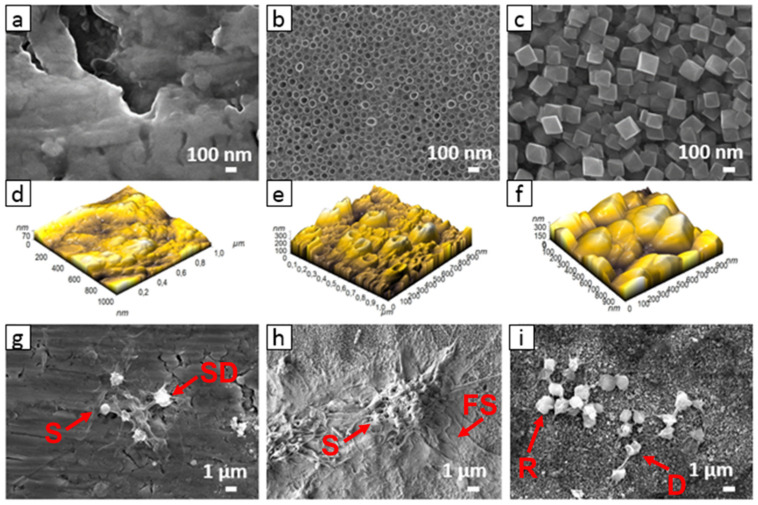
SEM and AFM images of Ti foil (**a**,**d**), TiO_2_ nanotubular surface (**b**,**e**), and TiO_2_ octahedral surface (**c**,**f**), and the adhered blood platelets on these three kinds of surfaces (**g**–**i**).

## 4. On the Role of Electrostatic Interactions

Electrostatic interactions between the charged surface and the electrolyte solution result in the formation of the electric double layer (EDL) [38,69,70,71,72,73,74,75,76,77]. In an EDL, ions with an electric charge of the opposite sign compared to the charged surface (counterions) are accumulated close to the charged surface, and the ions with a charge of the same sign as the surface (coions) are depleted from this region [69,70,71,78,79,80,81]. Due to a nonhomogeneous distribution of ions in the EDL, the electric field strength is screened at larger distances from the charged surface. Due to high magnitudes of the electric field in the EDL, the water dipoles near the charged surface are strongly oriented (Figure 7) [38,82,83,84,85,86,87,88,89,90].

In the past, the first theoretical description of the EDL was introduced by Helmholtz [91,92], who assumed that a single layer of counterions forms at the charged surface. Later, the spatial distribution of point-like ions in the vicinity of the charged surface was described by the Boltzmann distribution function for the counter-ions and co-ions in the Poisson equation [69,70] within the so-called classical Poisson–Boltzmann (PB) model [69,70]. The PB model neglects the finite size of molecules and considers the relative permittivity as a constant throughout the whole electrolyte solution, i.e., PB approach neglects the spatial dependence of relative permittivity. A constant relative permittivity is a relatively good approximation for small magnitudes of surface charge density, but not for higher magnitudes of surface charge density where a substantial decrease in relative permittivity due to the strong orientational ordering of water dipoles in the vicinity of the charged surface was predicted [38,86,88]. In addition, the PB model also does not take into account the spatially dependent volume electric charge distribution in the zwitterionic lipids headgroup region, which depends on the interaction with neighboring charged bodies and also on the composition of the electrolyte solution [38,39,43].

The finite size of ions in the theoretical description of the EDL was first incorporated by Stern [78] with the so-called distance of closest approach and later developed further by [71,79,80,81]. Their work was further improved by numerous theoretical studies and simulations taking into account the asymmetry of the size of the ions, direct interactions between ions, orientational ordering of water dipoles, discrete charge distribution of the surface, quantum mechanical approach, etc. [1,72,73,74,76,84,85,86,88,90,93,94,95,96,97,98,99,100,101,102,103,104,105,106,107,108,109,110,111,112,113,114,115]. The physical properties of the EDL are crucial in understanding the interaction between charged surfaces in electrolyte solutions [34,38,43,116,117,118,119,120,121,122,123,124].

### 4.1. Modified Langevin Poisson–Boltzmann model

In the following, we shall describe the theoretical consideration of the electrostatic interaction (adhesion) between lipid head groups of the proximal leaflet of the lipid bilayer and charged solid surface where the orientational degree of freedom of lipid headgroups is taken into account. Among others, we shall derive, within the modified Langevin Poisson–Boltzmann [38,125,126] model, an analytical expression for the osmotic pressure between two charged surfaces, which can then also be used for the calculation of osmotic pressure between the planar lipid bilayer and charged planar surface.

We shall start with a short description of the modified Langevin Poisson–Boltzmann (LPB) model of the electric double layer [38,125,126], which presents the generalization of classic Poisson–Boltzmann (PB) theory for point-like ions by taking into account the orientational ordering of water molecules in an EDL. In the modified LPB model, the orientational ordering of water dipoles is also considered close to the saturation regime or in the saturation regime, which leads to the prediction that the relative permittivity close to the charged surface is considerably reduced. The modified LPB model also accounts for the electronic polarization of the water [38,126]. The space dependency of the relative permittivity within the modified LPB model is [38,43,126]:(1)$$\varepsilon_{r}\left(r\right)=n^{2}+\frac{n_{w}p_{0}}{\varepsilon_{0}}\left(\frac{2+n^{2}}{3}\right)\left(\frac{L\left(\gamma p_{0}E\left(r\right)\beta \right)}{E\left(r\right)}\right),$$
where n is the refractive index of water, $n_{w}$ is the bulk number density of water, $p_{0}$ is the magnitude of the dipole moment of a water molecule, L(u)=coth(u)−1/u is the Langevin function, $\gamma =\left(2+n^{2}\right)/2$, E(r) is the magnitude (absolute value) of the electric field strength, β=1/kT, and kT is the thermal energy. The above expression for the space dependency of the relative permittivity (Equation (1)) then appears in the modified LPB equation for electric potential ϕ [38,43,126]:(2)$$\nabla \cdot \left[\varepsilon_{0}\varepsilon_{r}\left(r\right)\nabla \right]=2e_{0}n_{0}\sin h\left(e_{0}\phi \left(r\right)\beta \right),$$
where we take into account the macroscopic (net) volume charge density of the electrolyte solution written in the form:(3)$$\rho \left(r\right)=e_{0}\;n_{+}\left(r\right)-e_{0}\;n_{-}\left(r\right)=-2e_{0}n_{0}\sinh\left(e_{0}\phi \left(r\right)\beta \right)$$
and Boltzmann distribution functions for the number densities of monovalent cations and anions:(4)$$n_{+}\left(r\right)=n_{0}\exp\left(-e_{0}\phi \left(r\right)\beta \right),\;n_{-}\left(r\right)=n_{0}\left(e_{0}\phi \left(r\right)\beta \right)$$
where $n_{0}$ is the bulk number density of ions. In the limit of vanishing electric field strength, the above expression for the relative permittivity (Equation (1)) yields the Onsager limit expression for bulk relative permittivity [38,43,82]:(5)$$\varepsilon_{r,b}=n^{2}+{\left(\frac{2+n^{2}}{3}\right)}^{2}\frac{n_{w}p_{0}^{2}\beta }{2\varepsilon_{0}}$$

At room temperature T=298 K, $p_{0}=3.1$ Debye (the Debye is 3.336 × 10^−30^ C/m), and $n_{w}/N_{A}=55\;\mathrm{mol}/L$, Equation (5) gives $\varepsilon_{r,b}=78.5$ for bulk solution. The value $p_{0}=3.1\;\mathrm{Debye}$ is smaller than the corresponding value in previous similar models of electric double layers, also considering orientational ordering of the water dipole ($p_{0}=4.86\;\mathrm{Debye}$) (see, for example, [86,127]), which did not take into account the cavity field and electronic polarizability of water molecules. In addition, the model [127] predicts the increase in the relative permittivity in the direction toward the charged surface contrary to the prediction of the modified LPB model, which predicts the decrease in relative permittivity in the electrolyte solution near the charged surface [38,43,82] in agreement with experimental results. The predicted substantial increase in relative permittivity near the charged surface in [127], therefore, opposes the experimental results and defies the common principles in physics [87,123,125,126].

### 4.2. Osmotic Pressure between Two Charged Surfaces within Modified Langevin Poisson–Boltzmann Model

In the following, we shall derive, within the modified LPB theory, the expression for osmotic pressure between two charged planar surfaces (see Figure 7). First, we shall rearrange the modified LPB equation (Equation (2)) into a planar geometry in the form [38,43,125]:(6)$$-\frac{d}{dx}\left[\varepsilon_{0}n^{2}\frac{d\phi }{dx}\right]-n_{0w}p_{0}\left(\frac{2+n^{2}}{3}\right)\frac{d}{dx}L\left(\gamma p_{0}E\left(x\right)\beta \right)+2e_{0}n_{0}\sin h\left(e_{0}\phi \beta \right)=0,$$
where we take into account Equation (1) for relative permittivity. Equation (6) is first multiplied by $\phi^{\prime }=d\phi /dx$ and then integrated to obtain [38,43]
(7)$$-\frac{1}{2}\varepsilon_{0}n^{2}E{\left(x\right)}^{2}+2n_{0}kT\cos h\left(-e_{0}\phi \beta \right)-n_{w}p_{0}\left(\frac{2+n^{2}}{3}\right)E\left(x\right)L\left(\gamma p_{0}E\left(x\right)\beta \right)\;+\left(\frac{2+n^{2}}{3}\right)\frac{n_{w}}{\gamma \beta }\ln\left[\frac{\sin h(\gamma p_{0}E\left(x\right)\beta )}{\gamma p_{0}E\left(x\right)\beta }\right]=K,$$
where the constant K in Equation (7) is the local pressure between the charged surfaces. Equation (7) is equivalent to the contact theorem. In the second step, we subtract the bulk values (outside the space between the charged surfaces) from the local pressure between the charged surfaces to obtain the expression for the osmotic pressure difference $\Pi =\Pi_{inner}-\Pi_{bulk}$ in the form [38,43]:(8)$$\Pi =-\frac{1}{2}\varepsilon_{0}n^{2}E{\left(x\right)}^{2}+2n_{0}kT\left(\cos h\left(-e_{0}\phi \left(x\right)\beta \right)-1\right)\;-n_{w}p_{0}\left(\frac{2+n^{2}}{3}\right)E\left(x\right)L\left(\gamma p_{0}E\left(x\right)\beta \right)+\left(\frac{2+n^{2}}{3}\right)\frac{n_{w}}{\gamma \beta }\ln\left[\frac{\sin h(\gamma p_{0}E\left(x\right)\beta )}{\gamma p_{0}E\left(x\right)\beta }\right].$$

By taking into account Equation (4), we can rewrite Equation (8) in the form:(9)$$\Pi =-\frac{1}{2}\varepsilon_{0}n^{2}E{\left(x\right)}^{2}+kT\left(n_{+}\left(x\right)+n_{-}\left(x\right)-2n_{0}\right)-n_{w}p_{0}\left(\frac{2+n^{2}}{3}\right)E\left(x\right)L\left(\gamma p_{0}E\left(x\right)\beta \right)\;+\left(\frac{2+n^{2}}{3}\right)\frac{n_{w}}{\gamma \beta }\ln\left[\frac{\sin h(\gamma p_{0}E\left(x\right)\beta )}{\gamma p_{0}E\left(x\right)\beta }\right].$$

The osmotic pressure is constant everywhere in the solution between the charged plates (Figure 7).

If both surfaces have equal surface charge density ($\sigma_{1}=\sigma_{2}$), the electric field strength in the middle (x=H/2 in Figure 7) is zero; therefore, Equation (8) simplifies to the form [38]:(10)$$\Pi =2n_{0}kT\left(\cos h\left(-e_{0}\phi \left(x=H/2\right)\beta \right)-1\right).$$

**Figure 7 membranes-11-00533-f007:**
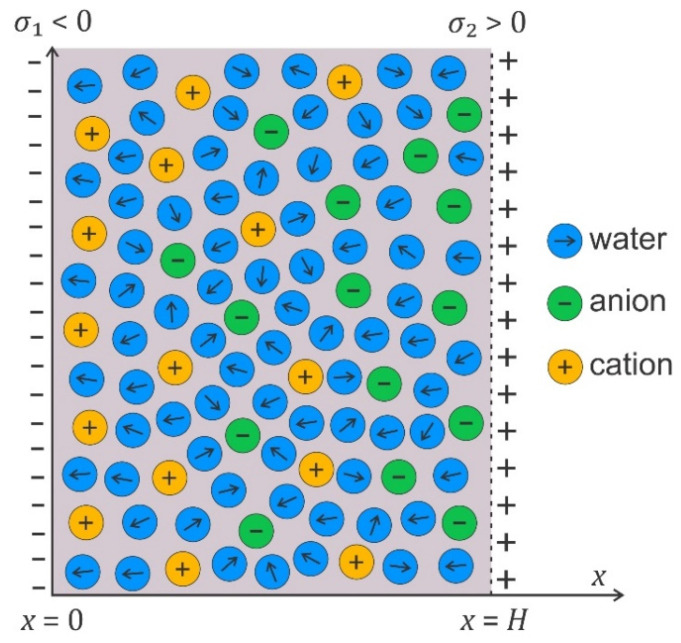
Schematic figure of an electrolyte solution between two charged surfaces at the distance H, where the surface charge densities $\sigma_{1}<0$ and $\sigma_{2}>0$.

For small values of $\gamma p_{0}E\left(x\right)\beta$ everywhere in the solution between the two charged surfaces, we expand the third and fourth term in the above Equation (9) into series to obtain:(11)$$\Pi \approx -\frac{1}{2}\varepsilon_{0}\left(n^{2}+{\left(\frac{2+n^{2}}{3}\right)}^{2}\frac{n_{w}p_{0}^{2}\beta }{2\varepsilon_{0}}\right)E{\left(x\right)}^{2}+kT\left(n_{+}\left(x\right)+n_{-}\left(x\right)-2n_{0}\right)\;=-\frac{1}{2}\varepsilon_{0}\varepsilon_{r,b}E{\left(x\right)}^{2}+kT\left(n_{+}\left(x\right)+n_{-}\left(x\right)-2n_{0}\right)\;=-\frac{1}{2}\varepsilon_{0}\varepsilon_{r,b}E{\left(x\right)}^{2}+2n_{0}kT\left(\cos h\left(-e_{0}\phi \beta \right)-1\right)\;,$$
where $\varepsilon_{r,b}$ is the Onsager expression for relative permittivity, defined by Equation (5). As, in thermodynamic equilibrium, the osmotic pressure is equal everywhere between the two charged surfaces (Figure 7), we can calculate the value of the magnitude of the electric field strength in Equation (10) also at the right charged surface (Figure 7) from the corresponding boundary condition, so Equation (11) then reads
(12)$$\Pi \approx -\frac{\sigma_{2}^{2}}{2\varepsilon_{0}\varepsilon_{r,b}}+2n_{0}kT\left(\cos h\left(-e_{0}\phi \left(x=H\right)\beta \right)-1\right),$$
where H is the distance between the two charged surfaces.

Figure 8 presents the osmotic pressure between negatively and positively charged flat surfaces as a function of the decreasing distance (H) between them, calculated within the modified LPB model.

### 4.3. Osmotic Pressure between Dipolar Zwitterionic Lipid Bilayer and Charged Rigid Surface

In the model, the zwitterionic dipolar lipid headgroup is composed of the lipids with a positively charged trimethylammonium group and a negatively charged carboxyl group, theoretically described by two charges at fixed distance, D (see Figure 9) [38,43,128]. The negative charges of the phosphate groups of dipolar (zwitterionic) lipids are described by negative surface charge density, $\sigma_{1}$ at x=0, while the opposite charged surface with surface charge density $\sigma_{2}$ is located at x=H. The corresponding Poisson equation in a planar geometry then reads [38,43,128]:(13)$$\frac{d}{dx}\left[\varepsilon_{0}\varepsilon_{r}\left(x\right)\frac{d\phi }{dx}\right]=2e_{0}n_{0}\sin h\left(e_{0}\phi \left(x\right)\beta \right)-\rho_{ZW}\left(x\right),$$
where $\rho_{ZW}\left(x\right)$ is the volume charge density due to the positively charged trimethylammonium group (Figure 9):(14)$$\rho_{ZW}\left(x\right)=\frac{\left|\sigma_{1}\right|P(x)}{D}\;\mathrm{and}\;\rho_{ZW}\left(x>D\right)=0,$$
and P(x) the probability density function [38,43,128]:(15)$$P(x)=\Lambda \frac{\alpha \exp(-e_{0}\phi \left(x\right)\beta )}{\alpha \exp(-e_{0}\phi \left(x\right)\beta )+1},\;0<x\leq D$$

The normalization constant is determined from the condition:(16)$$\;\frac{1}{D}\int_{0}^{D}P(x)dx=1.$$.

P(x) describes the probability that the positive charge of a dipolar lipid headgroup is located at the distance x from the negatively charged surface at x=0. The parameter α is equal to the ratio between the average volume of the positively charged parts of dipolar (zwitterionic) headgroups and the average volume of the salt solution in the headgroup region, meaning that the finite size of the positively charged part of the zwitterionic lipid headgroup is taken into account. The corresponding boundary conditions at x=0 and x=H should be taken into account [38]. The predictions of the model agree well with the results of MD simulations, as shown in [38,129].

To calculate the osmotic pressure between the zwitterionic headgroup region and positively charged surface (Figure 9), we can use Equation (8) with the input ϕ(x) and E(x) determined from Equations (13)–(16) at appropriate boundary conditions, where the values ϕ(x) and E(x) inserted in Equation (8) can be calculated for any D≤x≤H because, in thermodynamic equilibrium, the osmotic pressure is equal everywhere between the two charged surfaces. However, as we are using expression Equation (8), which neglects $\rho_{ZW}\left(x\right)$ (Equation (14)), we can calculate the osmotic pressure between the dipolar zwitterionic lipid bilayer and charged rigid surface (Figure 10) by using Equation (8) only in the region D≤x≤H, where $\rho_{ZW}\left(x\right)$ is different from zero.

**Figure 9 membranes-11-00533-f009:**
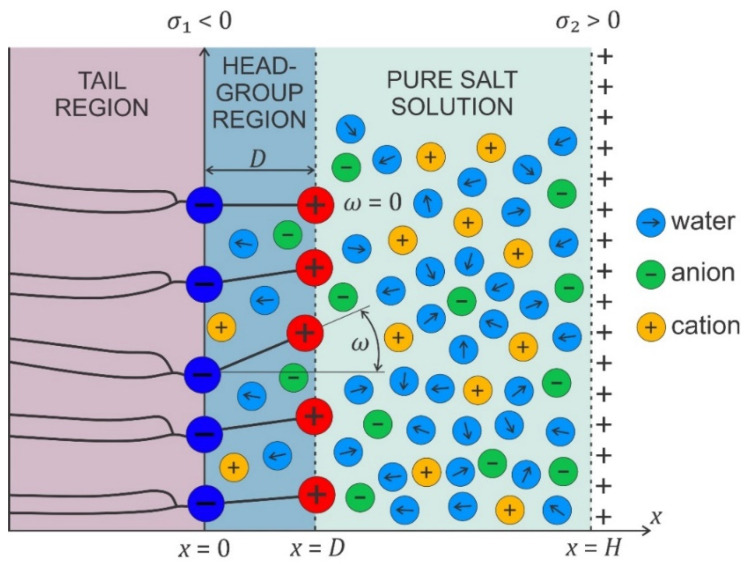
Schematic figure of the headgroup region composed of zwitterionic lipids with a positively charged trimethylammonium group and a negatively charged carboxyl group. The negative charges of the phosphate groups of the dipolar (zwitterionic) lipids are described by a negative surface charge density $\sigma_{1}$ at x=0, while the electric charge due to the positively charged trimethylammonium group is described by the spatially dependent volume charge density $\rho_{ZW}\left(x\right)$, defined in the region 0<x≤D (see Equations (14)–(16)). An example of a zwitterionic lipid is SOPC.

When the zwitterionic lipid layer approaches the negatively charged surface ($\sigma_{2}<0$), the average orientation of the lipid headgroup orientation angle (〈ω〉) decreases with decreasing H due to the electrostatic attraction between the positively charged parts of the lipid headgroups and the negatively charged surface, as schematically shown in Figure 11, based on the results presented in [38,43]. Accordingly, the osmotic pressure between the headgroups and the negatively charged surface decreases with decreasing H, as calculated in [38,43].

**Figure 10 membranes-11-00533-f010:**
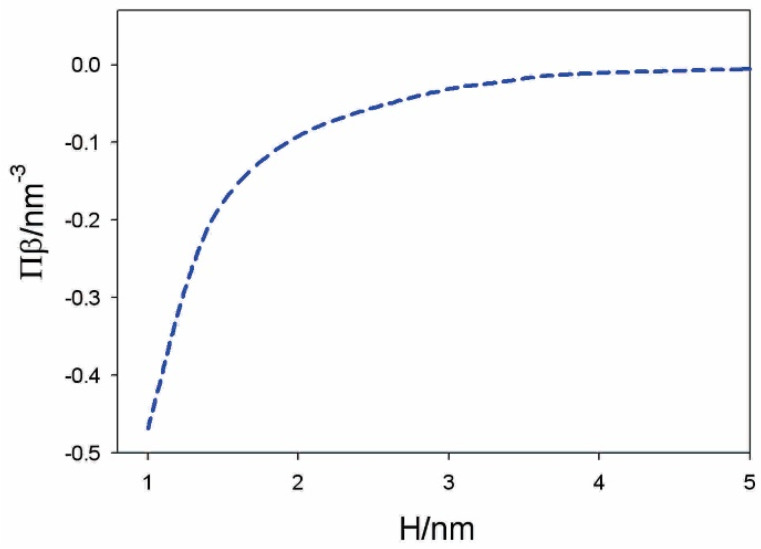
Calculated osmotic pressure between the dipolar headgroups and planar negatively charged surface as a function of the distance between the plane of the lipid phosphate groups and the charged surface (H) for alpha = 5. The values of model parameters are: T = 298 K, $\sigma_{1}=$ –0.30 As/m^2^, $\sigma_{2}=$ 0.30 As/m^2^, dipole moment of water $p_{0}=$ 3.1 Debye, bulk concentration of salt $n_{0}/N_{A}=$ 0.01 mol/L, and concentration of water $n_{w}/N_{A}=$ 55 mol/L. Reprinted from [38] with permission from Elsevier.

**Figure 11 membranes-11-00533-f011:**
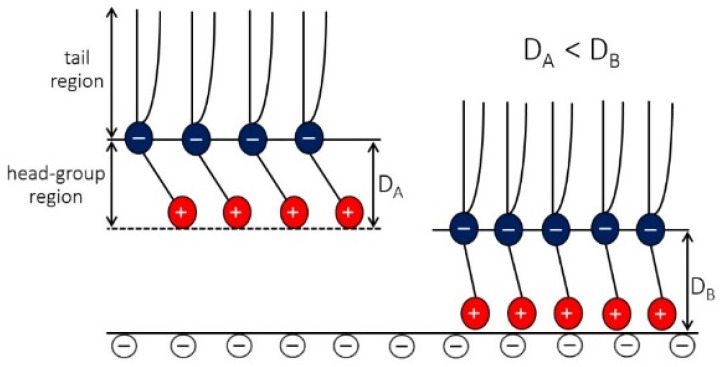
Schematic of the average orientation of the zwitterionic head-group at two different distances from the negatively charged surface. The figure, based on the results of theoretical modeling and MD simulations [38,128], shows that at smaller distances from the charged surface, the average orientation of the zwitterionic head-groups is more perpendicular to the charged surface.

## 5. Adhesion of Lipid Vesicles to Rigid Surface

In this section, we describe the interplay of membrane elasticity, geometrical constraints, and adhesive forces between the lipid bilayer and charged solid surface in the adsorption of lipid vesicles to the solid surface. The numerically calculated shapes of adhered lipid vesicles based on the system free-energy minimization are presented.

The shape of a vesicle upon adsorption to a surface is determined by the interplay of adhesion, bending, and geometrical constraints. This interplay is theoretically studied starting from a simple model in which the membrane experiences a contact potential arising from the attractive surface. Let us recall the free energy F expression of an adsorbed vesicle in terms of a simple model that takes into account the local bending energy terms, the adhesion energy and two geometrical constraints [130,131]:(17)$$F=\frac{1}{2}\kappa \oint {\left(C_{1}+C_{2}-C_{0}\right)}^{2}dA+\kappa_{G}\oint C_{1}C_{2}dA-WA_{c}+PV+\Sigma A,$$
where κ is the local bending modulus; $\kappa_{G}$ is the Gaussian curvature modulus; $C_{1}$, $C_{2}$, and $C_{0}$ denote the two principal curvatures and the (effective) spontaneous curvature, respectively; and dA is an infinitesimal membrane area element. In the third term, W is the strength of adhesion and $A_{c}$ is the contact area of the membrane and the surface. The last two terms represent the volume (V) and area (A) constraints with corresponding Lagrange multipliers P and
Σ. The normalization of the membrane free energy (Equation (17)) by the bending energy of a sphere for zero spontaneous curvature 8πκ leads to the expression for the reduced free energy $f=F_{b}/8\pi \kappa$:(18)$$f=\frac{1}{4}\oint {\left(c_{1}+c_{2}-c_{0}\right)}^{2}da-\frac{w}{2}\left(\frac{A_{c}}{A}\right)+p\oint dv+\sigma \oint da,$$
where $v=V/\left(4\pi R_{s}^{3}/3\right)$ is the reduced volume (see, for example, [131,132]); $a=A/4\pi R_{s}^{2}=1$ is the reduced area; $c_{0}=C_{0}R_{s}$, $c_{1}=C_{1}R_{s}$, and $c_{2}=C_{2}R_{s}$ are the reduced curvatures; p and σ are the reduced Lagrange multipliers; and
(19)$$w=WR_{s}^{2}/\kappa ,$$
is a dimensionless parameter, where $R_{s}=\sqrt{A/4\pi }$. The ratio $A_{c}/A=A_{c}/4\pi R_{s}^{2}$ is the reduced contact area and varies between zero for a spherical vesicle with reduced volume v=1.0 and 0.5 for pancake-shaped vesicles for a very small reduced (zero) volume v. The above energy expression (Equation (18)) is minimized numerically, as described in [133]. Note that, in Figure 12 (for $c_{0}=0$), the calculated nonadhered vesicle shapes corresponding to minimal bending energy and reduced volumes v≤0.591 are stomatocytic, while the shapes for 0.592≤v≤0.651 are oblate and, for v≥0.652, prolate (see also [134,135]). It can be seen in Figure 12 and Figure 13 that for high reduced adhesion strength w, the calculated shapes of adhered vesicles approach the limiting shapes composed of the sections of spheres corresponding to the maximal reduced contact area at a given reduced volume v.

## 6. Conclusions

The growing number of nanomaterial-based commercial products (medical implants, biosensors, antibacterial surfaces, cancer therapy, etc.) has generated an increasing need for thorough scientific studies to evaluate the interactions of nanomaterials with biological cells and the influence of these interactions on the stability and growing of cells. The cell membrane is a nanoscale barrier that protects the cell and constitutes the initial contact area with the nanostructured surface. The nanomaterial–membrane interactions are strongly dependent on the membrane curvature and is influenced by geometrical/topological constraints and mechanical and electrical properties of the membranes and nanostructured surfaces [1,2,22,47,49,53,54,62,66,67]. Hence, in the future, one of the major goals of the research will be to gain a deeper understanding into the mechanisms of nanomaterial–cell membrane interactions.

In this article, we describe briefly some selected experimental methods for studying the interactions between nanostructured surfaces and biological cells. As an example, we chose TiO_2_ nanotubular and octahedral surfaces and characterized them via AFM and XPS. The adhesion of human blood platelets to these surfaces was studied via SEM to elucidate the influence of the surface topography on the behavior of human cells adhered to TiO_2_ nanostructured surfaces.

In the theoretical part of the article, it is shown, among others, that the electrostatic interaction between the zwitterionic lipid head groups of the proximal leaflet of the lipid bilayer and the negatively charged solid surface is accompanied with a more perpendicular average orientation of the lipid head-groups. This may induce, among others, a more tightly packed gel phase of lipids in the adhered part of the vesicle membrane and stronger orientational ordering of water dipoles between the proximal lipid layer and the supporting solid surface. In the final part of the paper, we theoretically examined the influence of the vesicle size, bending modulus, and adhesion strength on the shapes of adhered lipid vesicles. The next step would be a thorough study of the changes in the thermodynamic properties such as the phase behavior of lipid bilayers associated with the lipid bilayer response to the supported nanostructured surface [62].

## Figures and Tables

**Figure 1 membranes-11-00533-f001:**
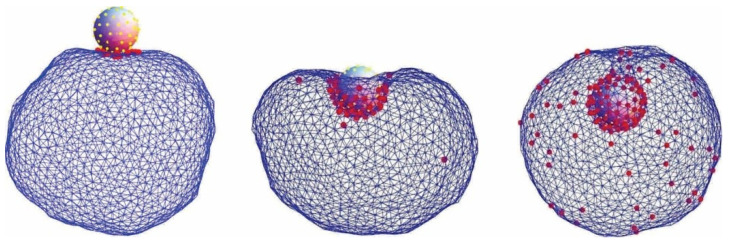
Encapsulation of a spherical charged particle macro-ion. Snapshots of representative configurations obtained from Monte Carlo simulations for different numbers of charged lipids (each having one unit charge) in the membrane: 15, 60, and 150 (from left to right). The right figure corresponds to the situation of nearly complete encapsulation of the macro-ion. The spherical particle carries 65 uniformly distributed, point-like cations of valence 2. Reprinted from [30] with permission of AIP Publishing.

**Figure 2 membranes-11-00533-f002:**
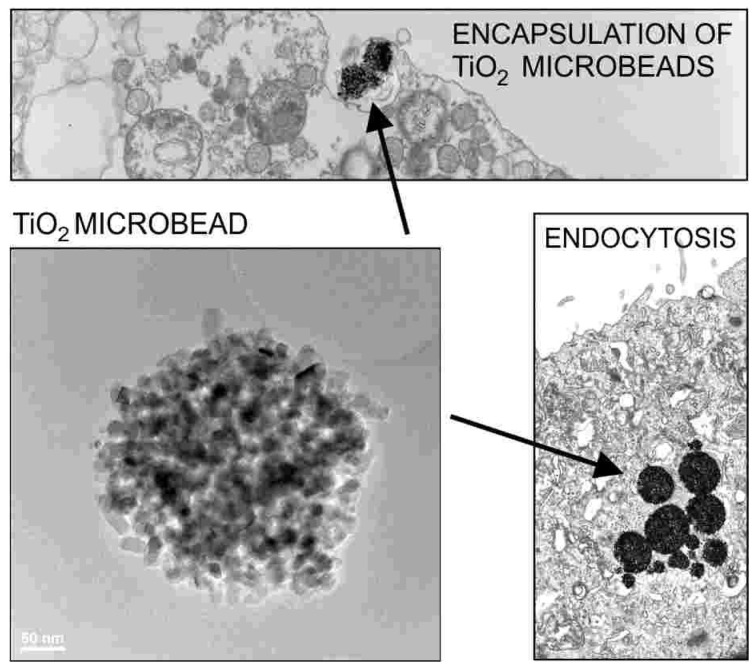
Encapsulation and endocytosis of TiO_2_ particles (microbeads) in the cells of urinary bladder (transmission and scanning electron microscopy). Reprinted from [32,33] with the permission of AIP Publishing.

**Figure 3 membranes-11-00533-f003:**
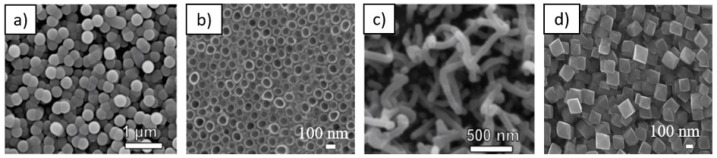
TiO_2_ nanostructured surfaces: (**a**) microbeads, (**b**) nanotubes, (**c**) nanowires and (**d**) octahedral nanoparticles. Partially reprinted (**a**,**c**) by permission from Springer Nature: Springer Protoplasma [51] (2016).

**Figure 8 membranes-11-00533-f008:**
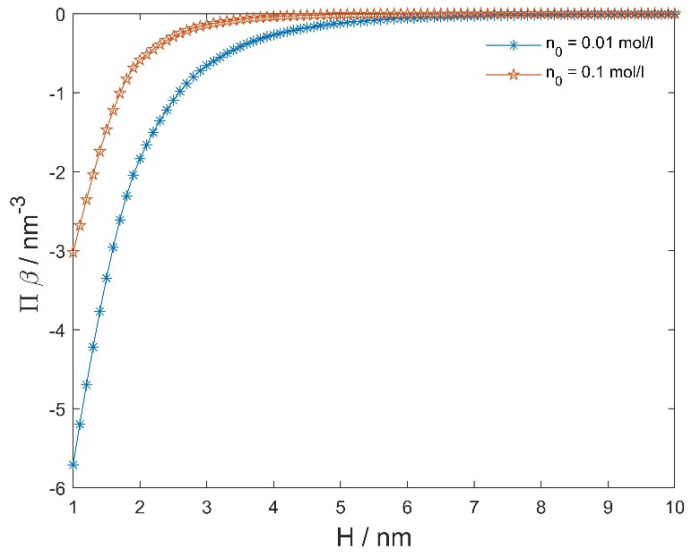
The calculated osmotic pressure between negatively and positively charged flat surfaces as a function of the distance between the two surfaces (H) (see Figure 7), calculated within the modified LPB model (Equations (1) and (2)) for two values of the bulk salt concentration. Other model parameters are: $\sigma_{1}=$ 0.2 As/m^2^, $\sigma_{2}=-\sigma_{1}$, T= 298 K, concentration of water $n_{w}/N_{A}=$ 55 mol/L, and dipole moment of water $p_{0}=$ 3.1 Debye, where $N_{A}$ is the Avogadro number.

**Figure 12 membranes-11-00533-f012:**
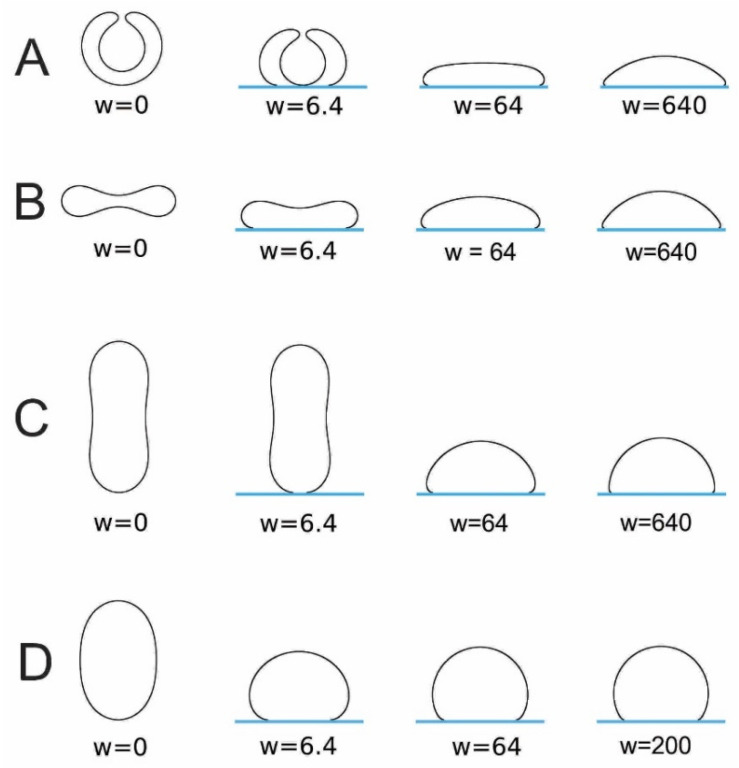
The calculated shapes of free (nonadsorbed) and adsorbed vesicles obtained by the minimization of the free energy given by Equation (18), determined for $c_{0}=0$ and different values of reduced volume: v=0.5 (**A**), 0.6 (**B**), 0.8 (**C**), and 0.95  (**D**), and different values of reduced adhesion strength w, defined by Equation (18).

**Figure 13 membranes-11-00533-f013:**
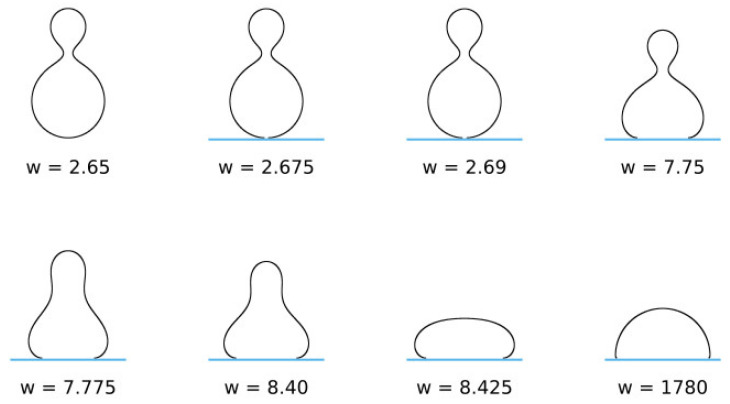
The calculated pear vesicle shapes of free (nonadsorbed) and adsorbed vesicles obtained by the minimization of the free energy given by Equation (18), determined for $c_{0}=$2.4, v=0.8, and different values of reduced adhesion strength w, defined by Equation (19).

**Table 1 membranes-11-00533-t001:** Atomic concentration (**%**) of detected elements by XPS on: Ti foil, hydrothermally treated Ti foil (HT), and TiO_2_ nanotubular (NT) surface (diameter of the nanotubes = 100 nm).

	O	C	Ti	F
**Ti foil**	37.1	51.6	11.3	/
**HT**	47.9	31.4	20.7	/
**NT**	40.2	37.9	16.5	5.4

## Data Availability

Not applicable.

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
