# Peer review of "Mechanical and Electrical Interaction of Biological Membranes with Nanoparticles and Nanostructured Surfaces"

_membranes, 2021, doi:10.3390/membranes11070533_

Round 1
Reviewer 1 Report
Please see my comments in the attachment

Reviewer 2 Report
In this review, the authors highlight the importance of the topography of nanomaterial surfaces and the role they play in their interactions with biological materials, using TiO2 as an example. They further go onto review the interplay of elastic and adhesive contributions of lipid vesicles to adsorption on these solid surfaces, using numerically predicted shapes of lipid vesicles. Finally they explore the origin of electrostatic interactions between lipid-bilayers and charged solid surfaces using established statistical mechanical approaches, and show that the electrostatic interaction between the zwitterionic lipid head-groups and the charged solid surface results in a perpendicular orientation of the lipid head-groups, leading to tighter packing of the lipids.
Overall, the review is well organized and written well. There are a few minor points, mostly related to the figures and typos.
The statistical mechanics part is beyond my expertise, so I have not looked into the details.
Most of the figure and table legends are lacking in details, for example :
- Table 1, what is O1, C1, F1?
- Figure 5, what is R, D, S, SD, etc.?
In Figure 6, it would be nice if the authors could indicate by arrows the morphological properties of the platelets which are discussed in the text, for broader readerships.
There are several typos statements which need to be re-phrased:
Line 139: typo: Nanopograhpy
Lines 246-249 (re-phrase)
Lines 295-298 (re-phrase)
Lines 435-438 (re-phrase)
The authors should describe in short the classic PB theory, before elaborating on LBP.
In Figure 9, the authors should provide examples of the different lipid head-groups (-ve & +ve).
In Figure 12, define the parameter w.
